# Application of a Prototype Thermoplastic Treatment Line in Order to Design a Thermal Treatment Process of Forgings with the Use of the Heat from the Forging Process

**DOI:** 10.3390/ma13112441

**Published:** 2020-05-27

**Authors:** Marek Hawryluk, Zbigniew Gronostajski, Maciej Zwierzchowski, Paweł Jabłoński, Artur Barełkowski, Jakub Krawczyk, Karol Jaśkiewicz, Marcin Rychlik

**Affiliations:** 1Department of Metal Forming, Welding and Metrology, Wroclaw University of Science and Technology, Lukasiewicz Street 5, 50-370 Wroclaw, Poland; zbigniew.gronostajski@pwr.edu.pl (Z.G.); maciej.zwierzchowski@pwr.edu.pl (M.Z.); pawel.jablonski@pwr.edu.pl (P.J.); artur.barelkowski@pwr.edu.pl (A.B.); jakub.krawczyk@pwr.edu.pl (J.K.); karol.jaskiewicz@pwr.edu.pl (K.J.); 2Kuźnia Jawor S.A., Kuziennicza 4, 50-370 Jawor, Poland; marcinrychlik@kuznia.com.pl

**Keywords:** thermoplastic treatment technology, controlled cooling line, heat forging, thermal treatment design

## Abstract

The global production of die forgings is an important branch of the motor industry for obvious reasons, resulting from the very good mechanical properties of the forged products. The expectations of the recipients, beside the implementation of the forging process, include also a range of supplementary procedures, such as finishing treatment including shot blasting, thermal treatment, and machining, in order to ensure the proper quality of the provided semi-product or the ready detail for the assembly line. Especially important in the aspect of the operational properties of the products is the thermal treatment of the forgings, which can be implemented in many variants, depending on the expected results. Unfortunately, a treatment of this type, realized separately after the forging process, is very time and energy-consuming; additionally, it significantly raises the production costs due to the increased energy consumption resulting from the necessity of repeated heating of the forgings for such thermal treatment. The article reviews the most frequently applied (separately, after the forging process) thermal treatments for die forgings together with the devices/lines assigned for them, as well as presents an alternative (thermoplastic) method of forging production with the use of the forging heat. The paper also presents a prototype semi-industrial controlled cooling line developed by the authors, which allows the development of the assumed heat treatment of forgings directly after forging with the use of forging heat, together with sample results of conducted tests.

## 1. Introduction

Hot die forging, despite being known for a long time now, is one of the most difficult production processes [1]. Currently, in addition to obtaining the assumed geometry of forged products, matrix forgings should have specific mechanical and plastic properties, which are obtained through heat treatment processes [2,3]. Since the mechanical properties are a reflection/dependent on the microstructure, therefore, it is necessary to recognize and analyze these relationships between the forging process and microstructure. The study [4] investigates the effect of various thermal treatment conditions on the microstructural evolution and correlates it with the mechanical properties obtained in the tensile strength test. The thermal treatment of die forgings is realized after the process of forging and trimming in order to unify the microstructure and properties of the forgings [5,6]. Depending on the requirements for forgings, such as the chemical composition, after the forging process, annealing, hardening, and tempering processes as well as oversaturation and aging are usually carried out. In fact, since the 1960s, attempts have been made to combine the plastic deformation process with heat treatment, because in many cases, this provides a synergistic combination of the effects of dynamic processes associated with hot deformation and transformations through diffusion. A suitable combination of both treatments (metal forming and thermal) allows for the mechanical properties of forgings, which are often greater than those that can be obtained by them separately.

In traditional die forging technologies, the high operational properties of the forgings are obtained through the selection of the steel with a proper content of carbon and alloy elements (unless the recipient demands a specific material grade) as well as the application of the toughening procedure. A typical cycle of the conventional thermal treatment of die forgings includes: heating of the charge to the initial forging temperature, i.e., about 1100–1300 °C, multi-step forging on a forging aggregate (press, hammer, etc.); the forgings after the forging process have the temperature of about 1100 °C and, depending on the technology, they can be hot trimmed; next, they are cooled in a box or sometimes on a controlled cooling line B-Y to ambient temperature; next, they are re-heated for normalization and cooled in air or heated to the austenitization temperature and cooled in oil, water, or another medium. The last procedures include tempering in the case of hardening, that is, re-heating the forgings to 550–650 °C and cooling them to the ambient temperature, straightening (calibrating) the forgings after hardening, and finishing relief annealing [7]. As it can be inferred, obtaining forgings with high strength properties as well as ductility as a result of separate forging process and next the conventional thermal treatment, requires additional operations related to heating within the given thermal treatment. This is connected with a prolonged production cycle and the accompanying additional costs of labor, device exploitation, cooling agents, and above all, the energy necessary for the repeated heating. In economically developed countries, the economic factors have enforced a reduction of the production time as well as energy consumption through the elimination of some of the mentioned thermal treatment operations. That is why one of the alternatives has become thermoplastic treatment, which takes advantage of the heat generated by the forging process, which in consequence eliminates the necessity of cooling the forgings after the forging process to the ambient temperature and their re-heating [8]. Typically, thermomechanical treatment is to change the mechanical properties of metals in the solid state (and as a result cause microstructural changes in them), which are the result of the combined action of temperature, time, and plastic deformation (the combined processes of heat treatment and plastic deformation). The application of thermoplastic processing technology needs to develop know-how joining knowledge in the field of kinetics of phase transformations during and after deformation with the technical possibilities of its application in industrial conditions. That is why the investigations performed in this field are fully justified, as they will provide many advantages, both in the scientific aspect and the financial benefits for obvious reasons.

## 2. Materials and Methods

The main aim of work is an overview of the currently used heat treatment of die forgings and the possibility of the application of heat directly from the forging process to design more efficiency in thermoplastic technology. Based on it, as a novelty, it presented its own research related to the application of a prototype-controlled cooling line in order to design the three different thermal treatment process of forgings (often carried out separately in industry) with the use of the heat from the forging process.

The research was divided into two stages:Carrying out various heat treatment variants in industrial conditions for a forked type forgings made of C45 steel (used in the steering gears of passenger cars). The tests were carried out to analyze such possibilities and to determine the structures and hardness obtained and to show the benefits of using forging heat for direct heat treatment.The concept of the prototype line for the design of different variants of forgings heat treatment with the possibility of the application of the forging heat was presented. Then, tests were carried out, in particular on the developed controlled cooling line for samples of two materials (C45 and 16MnCr5) used for the forging of yoke type, in order to show the line’s possibilities and verification as well as its use in industrial conditions.

## 3. Subject Matter

The thermal treatment technologies can be implemented by means of furnaces of different constructions, sizes, and power supply types, working in various cycles, atmospheres, and at different temperatures. In view of such diversity, it seems important to select the key parameters ensuring an efficient thermal treatment, in which the die forgings can obtain the proper structure and properties. This is especially important in the case of elements with complex geometry, which are problematic in their treatment, and their thermal treatment process is hard to control [9,10,11]. The most frequently expected structure is the temper sorbite obtained in steel forgings as a result of toughening through hardening and tempering. For the thermal treatment of forgings, chamber, pusher, and belt furnaces powered by gas burners or electric heaters are used. Much better effects are obtained on a technological line applying belt furnaces, on which the thermal treatment runs in a nitrogen shield. Slightly worse effects in respect of the forging quality (due to decarburization) but with similar efficiency were obtained in chamber furnaces. The selection of the thermal treatment technology should be determined mainly by the quality of the obtained microstructure and properties of the forgings. In the second place, one should take into the account the process efficiency, the possibility of its automatization, and thus the efficiency of the production process itself. The availability as well as the possibility of implementing thermal treatment procedures have been included in Figure 1. At present, among those mentioned in Figure 1, the most commonly used types of thermal treatment implemented at a large scale in the production of steel forgings are normalization, toughening, and isothermal annealing, which are usually performed separately from the forging process [12,13,14,15].

For stainless and austenitic steels, oversaturation is applied. For aluminum forgings, oversaturation and aging are usually performed. In the case of copper alloys, the most frequently applied procedures are oversaturation and aging [16]. In turn, for mono-phase titanium alpha alloys, partial annealing at lower temperatures is implemented as well as full annealing at higher temperatures. For pseudo-alpha alloys, normalization is applied. Diphase titanium alloys can be subjected to normalization as well as aging and tempering. Titanium beta alloys undergo toughening [17]. Interesting research using thermomechanical treatment for TiAl titanium alloy. The canned-forging and subsequent heat treatments of Ti-43Al-4Nb-1.5Mo alloy have been conducted, during which the hot deformation behavior, flow softening mechanism, microstructure evolution, and mechanical properties were investigated [18]. In turn, for magnesium alloy forgings, oversaturation as well as artificial and natural aging are applied. In addition, less often, recrystallization and relief annealing are implemented [19,20,21,22,23]. In the case of nickel alloys, heat treatment is primarily used to shape the properties of carbides prior to putting these alloys into service, ensuring that their desired structures and morphology are obtained. Depending on the chemical composition, requirements for operational properties and anticipated operating conditions, nickel alloys can be subjected to six basic types of heat treatment: normalization, homogenization, stress relief annealing, stress equalization, oversaturation, and precipitation hardening [24].

In the implementation of a thermal treatment process, the key device is the furnace in which the process takes place [25,26,27], and as well as conditions of heat exchange [28,29,30,31,32,33]. In this aspect, we distinguish between chamber, tunnel, pusher, and belt furnaces, as well as others, with special constructions and applications [34,35,36,37]. Beside the furnace, the technological line includes additional instrumentation, such as cooling lines [38], quenching tanks, washers, shot blasting machines, feeders, conveyors, trolleys, and platforms [39,40]. The whole stand or line for thermal treatment can be organized in different ways, depending on the needs and possibilities of the forging producer. A popular solution is the application of a chamber furnace, or a few furnaces placed next to each other in one row together with the quenching tanks and washers. The whole is operated by an industrial car, which makes it possible to transport the charge between the particular devices. A characteristic feature of such stands is the operation and treatment of the whole charge, which is placed on one pallet and then heated, cooled, and washed as a whole. This causes certain difficulties connected with the thermal inertia of the system, which is quite hard to heat and even harder to cool at a sufficiently high rate. An example of such a stand has been shown in Figure 2a. Another solution is a technological line in which the product is transported in a continuous manner lengthwise on the platform or the belt. Then, the particular elements of the stand (furnaces, tanks, washers) are placed in one line and connected to conveyors and feeders. In solutions of this kind, each detail (in belt furnaces) or a small portion of the product (in pusher furnaces) undergoes the treatment in an individual manner, which eliminates the effect of collectivity and the problems connected with transporting a large charge. An example of such a solution has been shown in Figure 2b.

This is connected with a prolonged production cycle as well as the accompanying costs of labor, operation and exploitation of devices, cooling agents, and most of all energy consumption, which is needed for the heating procedures, and also with environment pollution. Currently, in order to eliminate the costly and time-consuming toughening process, we can observe a development in the field of designing chemical compositions and technologies of die forging production, which takes place in four grade groups: pearlitic–ferritic steels, bainitic steels, multi-phase steels with a ferritic–bainitic–pearlitic structure and steels with a dislocation martensite structure. Another observed direction of die forging development is the group of steels whose chemical composition and manner of cooling after hot forging make it possible to obtain a product with a bainitic structure: lower, fine-grained, and non-carbide. A condition for the obtaining of a structure of this type is ensuring a fine-grained austenite structure after the end of deformation and a very strict control of the cooling process after the deformation [6]. So, an alternative method of producing die forgings with the use of toughening makes it possible to limit the production costs and reduce the energy consumption, which maintains the required properties of the forgings; this method is a combination of regulated accelerated cooling directly after the forging process. This means the use of the so-called forging heat and no necessity of cooling the forgings down to the ambient temperature and then re-heating them to the required temperature (high-temperature thermoplastic treatment). Research on the use of thermoplastic technology is being carried out more and more often, which aims to compare the obtained mechanical properties of forged products and the economics of such technology with current methods (separate forging and subsequent heat treatment of forgings). The study [41] compares the mechanical properties of metal sheets with a ferritic–martensitic structure made of steel C-Mn with micro-additions Nb and Ti produced by means of an energy saving thermoplastic treatment technology, with those obtained after the conventional hardening from a temperature above A_C1_. Such studies were also verified in [42], which deals with the impact of the use of ThermoPlastic Treatment (TPT) through forging on the structure and mechanical properties of micro-alloy steel. Forgings produced by this method were subjected to tempering in the temperature range from 550 to 650 °C, which resulted in obtaining the properties: Rp0.2, in the range of 925–993 MPa; Rm, in the range of 978–1061 MPa. In turn, the article [43] examined the impact of selected heat treatment parameters on the mechanical properties of a steam turbine blade operating at a temperature of up to 700 °C.

In addition, nowadays, together with the development of new steel grades, thermo plastic treatment is also increasingly used, due to the fact that such steels contain alloying elements that predispose this type of treatment. One of its applications is the use of such a solution for Fe-C-Al-based alloys that would make it possible to design high-strength construction steel not containing elements regulated as critical by the UE (European Union), such as Nb, V, and Mo. The work [44] showed that by using TPT (thermo plastic treatment), it is possible to obtain properties of alloys that do not contain critical elements that are comparable with the mechanical properties of traditional alloys. In addition, TPT is also used for other Ti [45] and Al [46]-based alloys. The mentioned studies also present a research-based confirmation of the economical and mechanical advantages of the method. In the available literature on the subject, you can find many other interesting studies on TPT, in which the authors used numerical modeling [47] regarding microstructural changes after different TPT variants [48], as well as the impact of changes in heating and cooling parameters immediately after forging on selected mechanical properties of forgings, including in [49,50,51] for the aviation industry.

## 4. The Die Forging Process with Selected Heat Treatment Variants

Figure 3 shows an example of a typical die-forging process of forked forgings, followed by a separate heat treatment of forgings. In the case of such a traditional forging process (Figure 3a), after forging is followed controlled cooling of the forgings on the cooling line (free cooling) where at the end of the procedure, the forgings fall into the container; their temperature is then around 610 °C (Figure 3b). It is close to the transformation temperature for this material: C45 steel (1.0503 according to DIN standard).

The forgings, after reaching the ambient temperature in a box, are transported to shot blasting and after it go to oil hardening, cooling, and tempering. The duration of the entire production cycle and that forgings are heated twice (during quenching and tempering) should be highlighted, which results in higher energy and refrigerant consumption (oil or polymer). On this basis, as part of the study, a decision was made to try to use heat from the forging process, and then carry out research work for three different heat treatment variants immediately after the forging process. Figure 4 presents the paths of the performed investigations for the following: oil hardening directly from the forging (marked in blue), oil hardening after the forging with special annealing at 870 °C (marked in gray), cooling down from the forging temperature 1200 to 600 °C (temperature below the transformation of perlitic), re-heating and annealing at 870 °C, and the next oil hardening (orange course). For all the three chosen variants, after oil hardening, the tempering treatment was performed (according for this forging). The assumed levels of temperatures were determined based off of the analysis for this material, i.e., steel C45, as well as the carried out investigations. For C45 steel, the perlitic transformation with free cooling takes place at about 660 °C (therefore, the forgings were cooled down to 600 °C), while the temperature of austenitization is in level of 860–880 °C.

Then, for all implemented variants (one forging was randomly selected from each), SEM (scanning electron microscope) microstructure analysis and microhardness measurements were performed. The samples for structural tests were etched with 3.5% Nital reagent (Figure 5). Hardness measurements were carried out using the Vickers method at 10 N load (Figure 6). Figure 5a shows a typical structure for tempered martensite with numerous fine precipitation of carbides (Fe_3_C) in a spherical form. Of course, there are also elongated forms of these precipitates. When comparing the microstructure for forgings made in accordance with Option 2, it differs significantly from Option 1 (Figure 5b). One can notice the precipitation of ferrite at the boundaries of ancient austenite grains, which indicates a typical sorbit structure after quenching. In SEM images, it is visible as darker areas that do not contain precipitates.

Figure 5c shows the results of microstructural tests for forgings made in accordance with the path for variant 3. In this case, you can also observe ferrite precipitation at the former austenite grain boundaries (dimensions 30–50 µm), as well as fine inclusions of Fe_3_C carbides in a spherical and elongated form. The results presented above in the hardening process of forgings made of C45 steel (microstructure, hardness) with the time–temperature parameters presented confirm that by correctly selecting them, the properties required by the final recipient can be obtained. What’s more, depending on the expectations of the recipient by changing these parameters, you can adjust the properties of the finished product. A comprehensive heat treatment was carried out for each variant. The treatment after the hardening process enters the process of tempering. Therefore, there are very small differences in the structure of the material.

You can see from the measurement results (Figure 5), basically regardless of the heat treatment variant used (from the proposed ones), the hardness values are in the range of 235–255 HV. Such values are also within the range according to the recipient’s requirements. Therefore, it can be concluded that the proposed heat treatment variants carried out directly from the forging process do not have a negative impact on both the microstructure and the hardness of the forgings obtained after separate/independent heat treatment. It seems that hardening directly from the forging temperature is possible, but from a technological point of view, it is unjustified. The high hardening stress generated can cause product deformation and/or cracking during the cooling process.

In addition, carrying out the proposed heat treatment variants directly from the forging temperature is of course economically justified, both in terms of process efficiency and production costs. It is obviously important to precisely define the cooling times and rates to ensure adequate temperature control over a specific range of times and temperatures

## 5. The Concept of the Prototype Line for the Design of Different Variants of Forgings Heat Treatment with Possibility the Application of the Forging Heat

To examine and analyze the impact of selected time–temperature parameters on the heat treatment of forgings, the authors developed and built a prototype experimental line to test different heat treatment variants. The line is based on the so-called high temperature thermoplastic treatment enabling the simulation and optimization of the thermal treatment of processes carried out in industrial conditions. Based on the research and analysis of the results obtained on the experimental line, an industrial line will then be designed and built. Ultimately, such an industrial line will enable the design and conduct of specific heat treatment in industrial conditions in die forges.

The concept of the prototype line is based on any (depending on the requirements of the developed thermal processing technology) arrangement of four key devices: mechanical press for hot and hot forging (P), controlled cooling line (L), chamber furnace for heating the charge/forgings (F), and oil/polymer hardening baths (Q). Thanks to this, through any configuration of devices, it will be possible to perform any thermoplastic treatment, including, e.g., both normalization and quenching (Figure 7). With the right selection of these key devices, it is possible to simulate almost any heat treatment process, both as a separate one and immediately after the forging process (Figure 8).

New in the proposed solution, i.e., the entire experimental line, is the prototype line for the controlled cooling of forgings. It is built of a chamber with a length of 3–4 m (with the possibility of attaching or removing shielding panels or adding one segment) and a width of 500 mm, together with a conveyor belt for transporting forgings/charge material.

Directly above the chamber there are 4 mechanical fans, each with a power of over 600 W, which allow you to pump air into the chamber and its circulation to cool the analyzed elements. The speed of the ventilators and the tape feed is regulated by means of frequency converters. Owing to the use of a high velocity ratio, it is possible to fully load the line with the weight of up to 150 kg/m. The side walls of the chamber were designed in such a way so that the biggest possible stream of air would flow around the forgings moving on the belt. An important aspect was such a construction design which would make it possible for the forging to be cooled uniformly on each side, including the bottom. For this reason, a conveyor belt made of a heat-resisting net with big meshes was used, which minimized the air resistance and reduced the ground effect.

### 5.1. Exemplary Tests under Semi-Industrial Conditions

In order to verify the developed line, preliminary tests were carried out on a semi-industrial stand using feedstock, i.e., two steels: 16MnCr5 (1.7131 according to DIN) and C45. Then, test samples with dimensions: ⌀35 × 20 mm and ⌀35 × 40 mm were prepared to further analyze the effect of volume changes. Twelve identical samples were made for each of both materials; for each geometry, i.e., 24 samples in total for each steel (12 with dimensions ⌀35 × 20 mm and 12 with dimensions ⌀35 × 40 mm).

Separate heating was used for each sample because of the possibility of scale formation in the case of repeated heating of the same element. Each test consisted of heating the sample material in an oven to a temperature just above 1100 °C and then subjecting it to a controlled cooling process to about 400 °C on a controlled cooling line. In addition, to analyze the impact of fan speeds, a different fan speed was used for each sample in a given series (dimensions and material), which was changed by 5 Hz using a frequency converter in the range from 0 Hz (which was 0 rpm) to 55 Hz (which was respectively 1650 rpm) for maximum fan speed. The belt speed for all measurements was constant at around 2 m/min. Preliminary tests showed that the change in belt speed had no significant effect on the sample cooling rate. The value of the belt speed was adopted in such a way that for the switched off fans, a single sample was cooled from the forging temperature from 400 °C. During each process, the temperature of the middle of the sample was recorded using a K-type thermocouple with a diameter of 1.2 mm. During each test, the cooling time was measured as the difference between the time when the sample with the forging temperature of 1100 °C reached 400 °C. In Figure 9 and Figure 10, the cooling processes for both materials (with dimensions 35 × 40 mm) were discussed.

The cooling temperature of the onset of the phase transformation of austenite into perlite can be observed as the cooling rate increases. Of course, it occurs much faster at higher cooling speeds. Comparing the results obtained for 16MnCr5 steel (Figure 9) with respect to C45 steel (Figure 10), phase transformation is less visible due to the higher carbon content and the lack of alloying elements.

Figure 11 presents the distribution for hardness measurements (as the average of three measurement points) for both materials: 16MnCr5 and C45 as a function of fan speed for samples with dimensions 35 × 20 mm, as mechanical properties.

For the graph presented in Figure 11, differences in hardness result from the material characteristics (higher carbon content: a higher amount of perlite in the material structure). The presence of perlite determines most of the hardness, as is the case with C45 steel. The differences in hardness between the two materials are around 30–40 HB. As the fan speed increases, the hardness increases for both materials. For customers who do not require the structure of martensite tempered in the forging, there is the possibility of using controlled cooling from the forging temperature. For yoke-type forgings made of both materials, the hardness should be in the range suitable for C45 (175–260 HB) steel, and for 16MnCr5 (150–240 HB). Hence, on this basis, it can be concluded that by choosing the appropriate fan speed (controlled cooling) on the developed line, it is possible to achieve the assumed hardness.

On the other hand, Figure 12 presents sets of cooling courses for two different materials and two cooling rates for the same, but only slightly smaller sample sizes (35 × 20 mm), because there were no significant differences in the waveforms for larger (35 × 40 mm) and smaller (35 × 20 mm) sample sizes of the same material.

After compiling any two cooling courses for samples with the same dimensions and cooling parameters, but for different materials, we can also notice that despite the more energetic transformations occurring in the material at C45, the cooling time is ultimately the same. With an increase of the cooling rate, the temperature of the transformation lowers. This is clearly visible for the material C45. It is an important parameter that should be taken into consideration while designing thermal treatment processes.

To be able to optimize the times of designed thermal treatments, which are important in terms of the production time, based on the collected data and measured cooling times as well as the dependence of cooling times on temperature, a fan speed diagram was developed (Figure 13). For example, the characteristics for the sample with dimensions ⌀35 × 40 mm made of C45 (Figure 13–light cyan line) were determined based on the times measured for a series of 12 such bars, with different speeds of fans (Figure 10).

Analyzing the obtained cooling curve runs, it can be seen that for the two different materials used, which are forged in the industrial process, the cooling times for the same sample geometry are similar (Figure 13). Some differences may be caused, e.g., by slightly different emissivities of materials, which causes different intensities of scale formation on samples. It can also be seen that individual runs are parallel, regardless of the geometry/volume of the samples.

### 5.2. The Initial Testing of the Yoke Forging Used in Car Steering Gears

First, cooling curves were determined during an industrial process on an industrial cooling line, referred to as B-Y (marked as dashed lines). On the other hand, cooling curves for the same forgings as in the forgings industry, heated in the furnace to 1100 °C and cooled to 400 °C were determined on the constructed cooling line. In addition, two variants were adopted on the prototype line for controlled cooling: the maximum fan speed was used for cooling a single forging and forging cooling (with a thermocouple) cooled on the line and placed between other heated forgings, as is the case in industry (Figure 14).

This was to determine the effect of cooling for a single forging and placed with other forgings (additional heat source), because all previous tests on the line were carried out on individual forgings. During each test, the temperature was recorded in the back of the forging (body) and in one of its arms (fork) using shielded K-type thermocouples (Figure 15) due to forging dimensional differences (volume difference between the fork and body).

As can be seen in Figure 14, for cooling forgings on an industrial BY line from forging temperature 1100 °C to 400 °C, the cooling time is about 600 s. While for a prototype line for controlled cooling, with maximum fan efficiency (1650 rpm), the cooling time was reduced by almost half to about 360 s. On the other hand, the difference between the measuring points (fork and body) is about 50 °C. In addition, the cooling time from forging temperature to 400 °C for a single forging placed between the remaining forgings increased by about 90 s.

The test results presented above on the developed controlled cooling line are preliminary but promising results. Therefore, conducting our research, as well as developing such devices is justified, especially in view of the indisputable use of thermoplastic treatment. As demonstrated by the selection of appropriate time–temperature parameters, it is possible to simulate almost any conditions that occur in industrial conditions, including die forges, where it should bring measurable financial benefits.

### 5.3. The Directions of Further Investigations

In the next stages of research, it is assumed that the semi-industrial stand for the design of thermoplastic treatment will be expanded, including, among others: controlled cooling line, with air flow velocity measurement systems. Due to this fact, a measurement in two axes was made: the longitudinal axis in respect of the feeder belt and the perpendicular axis (along the ventilator axis). The measurements were performed for the ventilator speed of 300 rpm to 1500 rpm, with the resolution every 150 rpm on the whole working length of the conveyor. The sensor was mounted on the conveyor belt and moved at the speed of 0.5 m/s, which was similar to the earlier cases. This will allow you to determine the temperature distribution and air flow inside line. On the basis of the presented diagrams (Figure 16 and Figure 17), we can see that the speed along the belt is much higher (average of about 4 m/s) than in the case of the measurement along the ventilator axis (about 2 m/s). This is caused by the ground effect, occurring as a result of accumulated air masses, which on meeting the resistance of the conveyor belt grid, escape through the sides of the device (entry and exit). 

An increased speed in the case of axis X can be observed in the areas of the central ventilators, whereas the lowest speed is observed between the ventilators. In the case of axis Y, the highest speed is observed between the ventilators and the lowest speed: in the passages through the very center of the ventilators.

It also schedules a comparative analysis of identical forged elements made of two different materials and next an examination of the effect of the performed thermal treatments on the mechanical properties. The future research will also include a cycle of thermal treatments performed on forgings and a comparison of their mechanical properties with the properties of forgings subjected to the traditional thermal treatment. It is also planned to use numerical modeling, e.g., using Marc and Abaqus computing packages for so-called CFD (computational fluid dynamics) analysis. Then, it will be possible to more accurately determine the air flow in the chamber, which will allow the cooling chamber to be designed in such a way that the generated air rotation will be used effectively.

## 6. Summary and Conclusions

This work presented an overview of currently used heat treatments with necessary equipment dedicated for the treatment of die forgings investigations. On this basis, it has been demonstrated that it is fully justified in many cases to perform thermal treatment directly using the heat generated during the forging process, without the need to cool to ambient temperature and reheat. As a result of using this type of treatment, a significant amount of energy is obtained, which significantly reduces the process time and reduces production costs. The results of the research show that the proper selection of time–temperature parameters (the use of controlled cooling lines) can obtain such mechanical properties of forgings that are even better than assumed, compared to traditional (separate) and time-consuming and expensive thermal treatment performed without using heat forging. Conducted research showed that for yoke-type forgings made of both materials, the hardness should be in the range suitable for C45 (180–260 HB) steel and for 16MnCr5 (150–240 HB). Based on it for customers who do not require the specific structure after being tempered in the forgings, there is the possibility of an application-developed controlled cooling line. Additionally, through the optimization of the construction and the consideration of the size and direction of the air flow in the cooling chamber of the develop cooling line, it also becomes possible to reduce the cost of manufacturing forgings with different geometries and specific operational properties.

## Figures and Tables

**Figure 1 materials-13-02441-f001:**
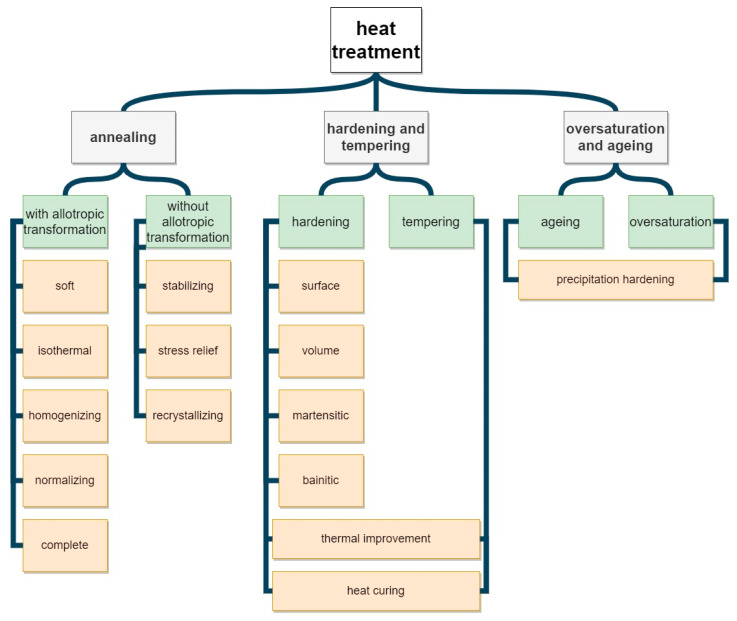
Division of the thermal treatment procedures of die forgings.

**Figure 2 materials-13-02441-f002:**
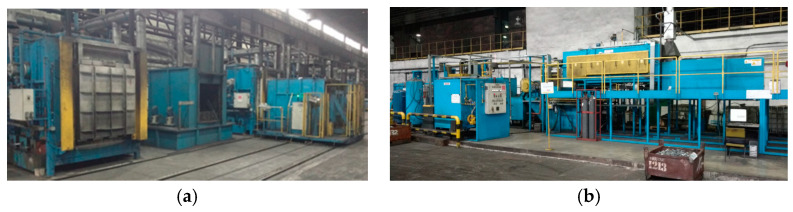
Exemplary images of (**a**) the stand for the thermal treatment of forgings in a chamber furnace and (**b**) the stand for the thermal treatment based on a belt furnace.

**Figure 3 materials-13-02441-f003:**
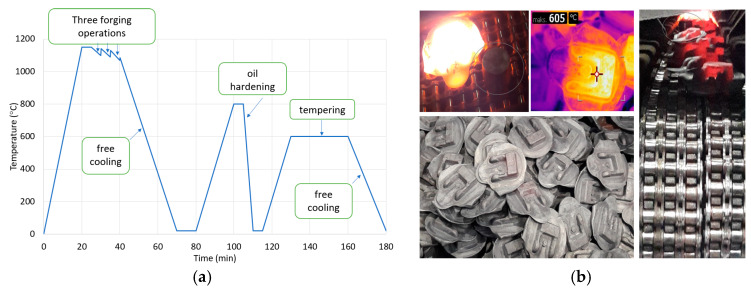
The views of (**a**) typical die-forging process an example with a separate heat treatment, (**b**) photos of forked forgings during and after a controlled cooling process.

**Figure 4 materials-13-02441-f004:**
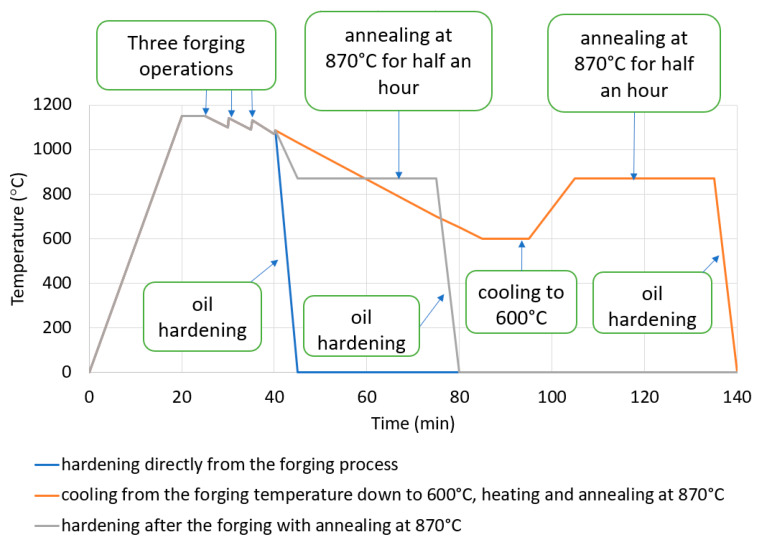
The paths scheme of the proposed thermal treatment variants directly after the forging process.

**Figure 5 materials-13-02441-f005:**
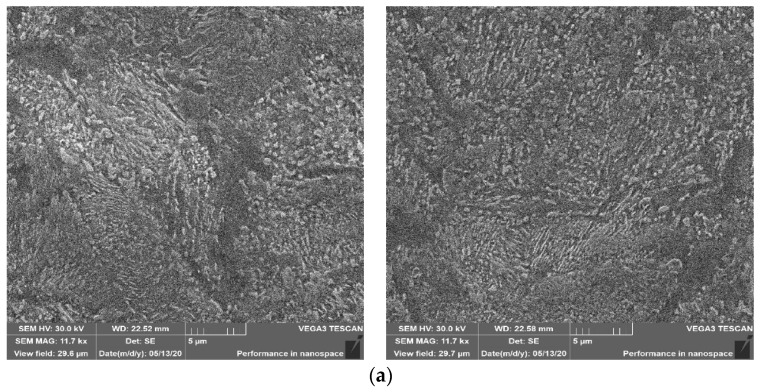
SEM scanning microscopic analysis after three proposed treatment variants (hardening) directly after the forging process: (**a**) technology I, (**b**) technology II, (**c**) technology III, and (**d**) the places of the metallographic specimens was sampled.

**Figure 6 materials-13-02441-f006:**
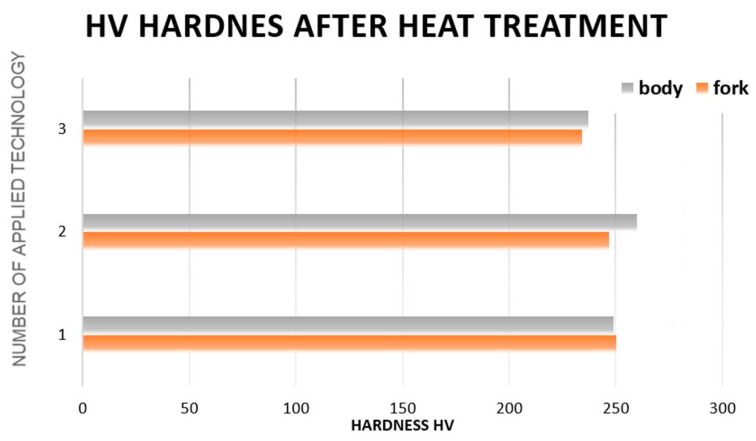
Microstructural test results for the three selected thermal treatment variants: thermal improvement.

**Figure 7 materials-13-02441-f007:**
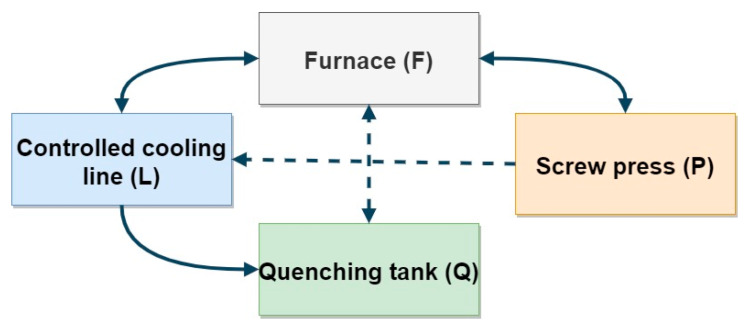
The flowchart of a test station for the processing of thermoplastic.

**Figure 8 materials-13-02441-f008:**
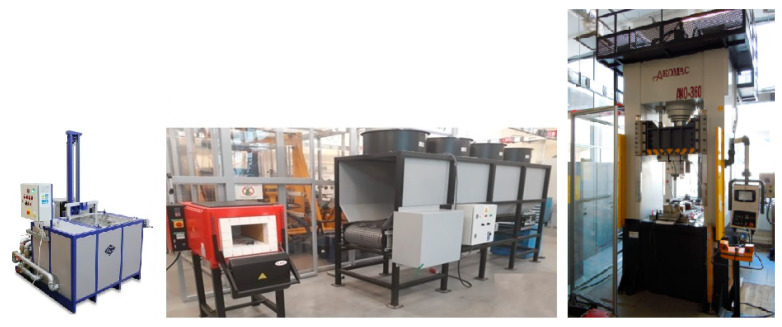
Exemplary arrangement of the test stand for thermoplastic treatment.

**Figure 9 materials-13-02441-f009:**
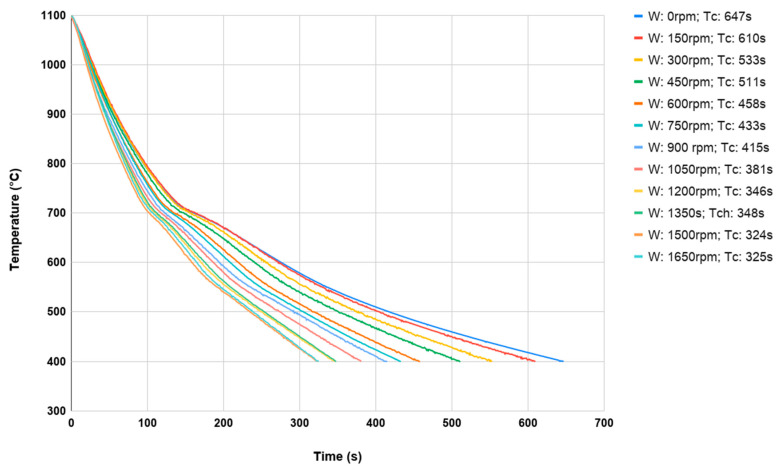
The cooling curves of a samples made of 16MnCr5 ⌀35 × 40 mm from forging temperature 1100 °C to 400 °C.

**Figure 10 materials-13-02441-f010:**
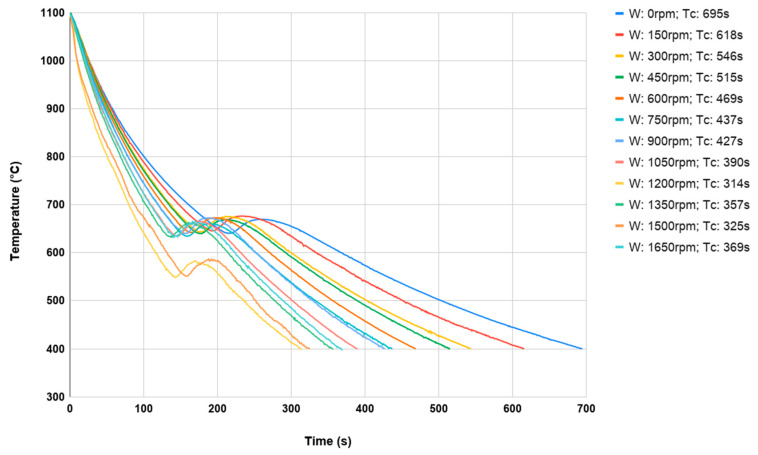
The cooling curves of a samples made of C45 35 × 40 mm from forging temperature 1100 °C to 400 °C.

**Figure 11 materials-13-02441-f011:**
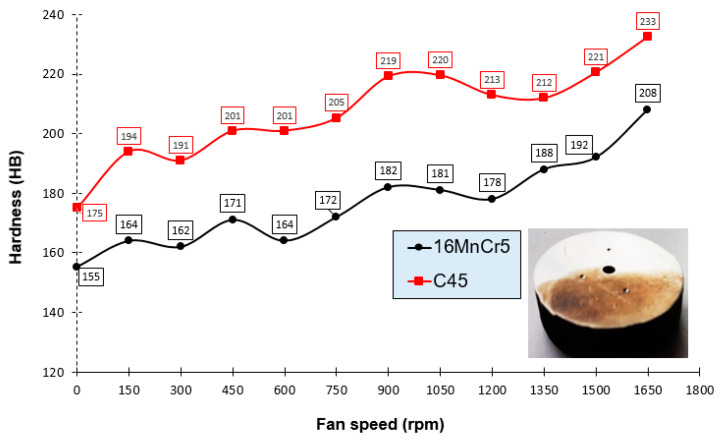
The distribution for hardness measurements for both materials (16MnCr5 and C45) as a function of fan speed for samples with dimensions 35 × 40 mm.

**Figure 12 materials-13-02441-f012:**
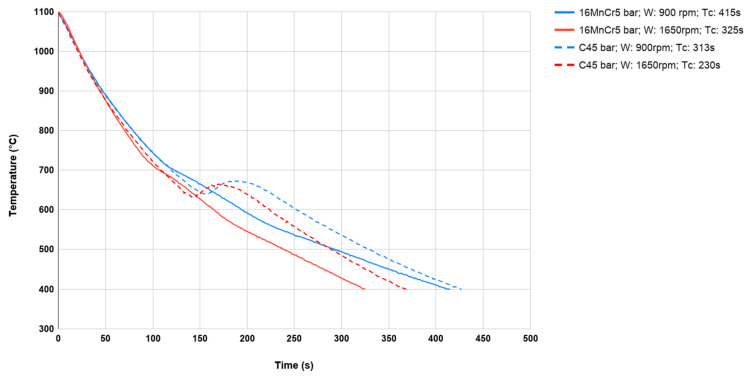
The cooling curves of 16MnCr5 and C45 samples with the dimensions ⌀35 × 20 mm with the ventilator speed of 900 rpm and 1650 rpm from forging temperature 1100 °C to 400 °C.

**Figure 13 materials-13-02441-f013:**
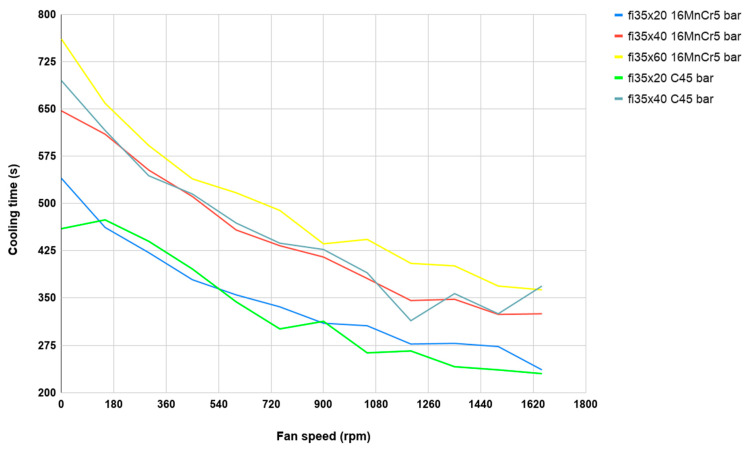
The cooling curves of 16MnCr5 and C45 samples with the dimensions ⌀35 × 20 mm, ⌀35 × 40 mm and ⌀35 × 60 mm, from forging temperature 1100 °C to 400 °C, in the function of the speed of ventilator.

**Figure 14 materials-13-02441-f014:**
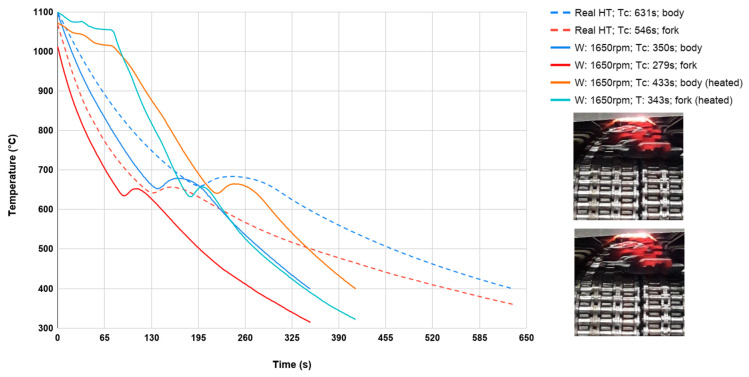
Cooling course of a yoke forging within the scope of 1100–400 °C in the function of time for different ventilator speeds (pictures from industrial cooling process).

**Figure 15 materials-13-02441-f015:**
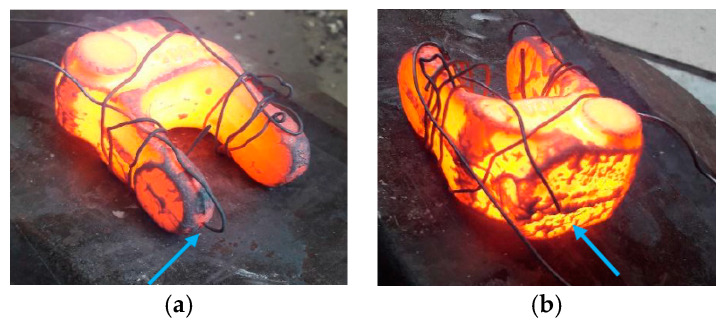
The forgings with the place of the thermocouples assembly marked on the prototype controlled cooling line: (**a**) in the fork and (**b**) in the body of forging.

**Figure 16 materials-13-02441-f016:**
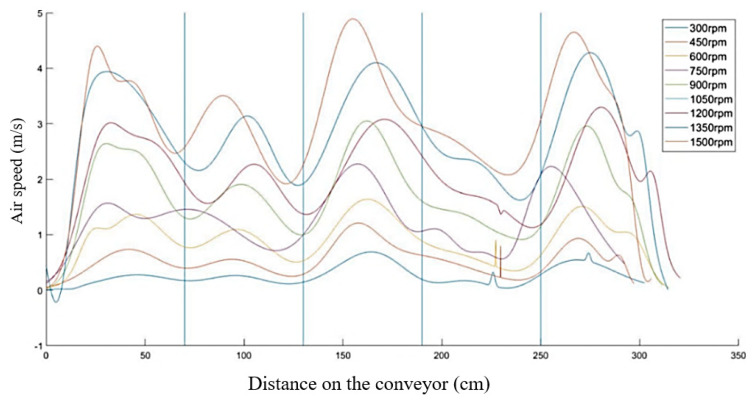
Distribution of the air flow speed in the scope of 300–1500 rpm along the ventilator axis: vertical lines (axis Y).

**Figure 17 materials-13-02441-f017:**
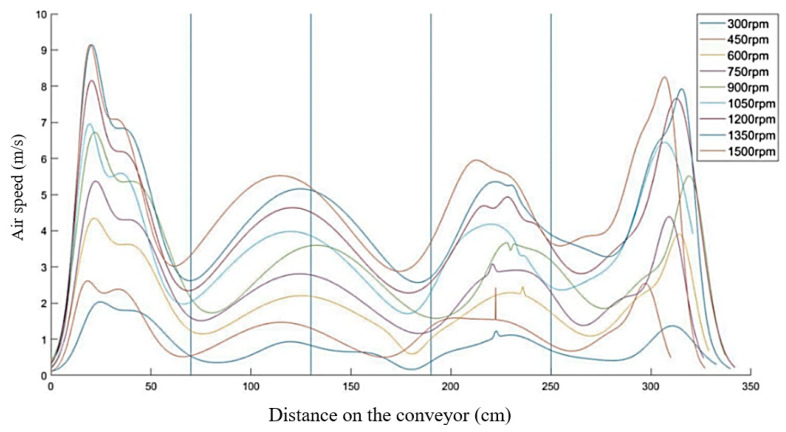
Distribution of the air flow speed in the scope of 300–1500 rpm along the conveyor belt shift axis (axis X); vertical lines: ventilators axes.

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
