# Peer review of "Application of a Prototype Thermoplastic Treatment Line in Order to Design a Thermal Treatment Process of Forgings with the Use of the Heat from the Forging Process"

_materials, 2020, doi:10.3390/ma13112441_

Round 1
Reviewer 1 Report
Dear Authors.
The research paper aims to perform thermal treatment using the heat generated during the forging process, without the need to cool to ambient temperature and reheat. It results in the process time and the production costs reduction. But the paper noticed to have many spelling mistakes and unexplained sentences like abbrevation, etc. The paper can be accepted for publication after the following changes.
- In introduction, in line 52, "A suitable combination of the two". Here what is mean by two? Is it adding any two different procedures together. Please explain it.
- in Line 131, spelling mistake : "transformation"
- in Line 142: looks repeating : "TiAl titanium alloy TiAl". correct it.
- in Lines: 188,197,211. References error.
- In Line 201: "TPT" add abbrevation.
- In Figure 6. "Hardness" "Applied". spelling mistakes
- In Figure 10. "Please explain why there was small oscillations in the curve (yellow line) around 150-200 seconds"
- Figure 14. Add subcaptions for better understanding.
- Lines 424-430: Add these lines into future wok title by adding new section.
- Please try to edit the figure into high resolution for better visuals. Also add more symbols to explain the curves rather than putting line colors.
Thank You.
Author Response
Dear Editor and Reviewers,
The authors would like to thank the editor and reviewers very much for sending the detailed reviews. The comments sent allowed for significant raise in the scientific level of the article. The most important changes in the corrected version of the manuscript have been highlighted in red, while the grammatical and stylistic changes were not marked in order to make the document more readable. The detailed answers for 1st reviewer`s remarks may be found in separate file.
regards,

Reviewer 2 Report
The paper presents very interesting information on the improvement of the efficiency of forging processes. It is quite well-written and original and, most importantly, scientifically sound. There are mostly problems with the presentation and some minor mistakes. The only scientific problem is the lack of microstructural characterization of the materials.
1- The descriptions in Figures are too small to be read, especially on A4 paper. Make them much larger and readable otherwise, there is no point in showing them. Some marks are really invisible, like those markings of the heat cycle in Fig. 3a.
2- There are some wrong terms being used, I believe they may have resulted from wrong translations; for instance, " hardening and tempering processes as well as supersaturation and aging" supersaturation in usually called solution annealing. At other places, authors call it "oversaturation"
3- The authors are using a lot of general terms and long sentences which are sometimes hard to follow. For example here:" forgings with high strength properties as well as ductility by way of a forging " what does it even mean by way of forging?? Kindly be more specific describe everything correctly.
4- There are many errors such as this: "[Error! Reference source not found.]." It should not be present at the paper at this point and it shows the carelessness of authors.
5- Two steels were used for the tests: 16MnCr5 and C45. However, the differences in cooling between these two steels are not explained, in relation to their microstructures and physical properties.
6- The microstructural characterization in Fig. 5 does not show enough details. It would be beneficial to show also SEM images and describe microstructure more quantitatively. Additionally, even though the hardnesses after different treatments are similar, the properties such as tensile strength of notched toughness can be vastly different. This should be discussed.
Author Response
Dear Editor and Reviewers,
The authors would like to thank the editor and reviewers very much for sending the detailed reviews. The comments sent allowed for significant raise in the scientific level of the article. The most important changes in the corrected version of the manuscript have been highlighted in red, while the grammatical and stylistic changes were not marked in order to make the document more readable. The detailed answers for 2nd reviewers’ remarks may be found in separate files.
regards,

Reviewer 3 Report
Review
The theme addressed in the paper is focused on application of a prototype thermo-plastic treatment line in order to design a thermal treatment process of forgings with the use of the heat from the forging process
From the analysis of the information presented in the article, I found the following:
- the introduction must be restructured so as to be more concise and direct references to the research topic analyzed;
- the research methodology is missing. For example, the "Materials and methods" section is completely missing;
- Sub-item 2 must be removed from the paper.
- The information presented in this subpoint is specific to a student course and not to a scientific paper;
- Figure 3 must have a bibliographic source or must be removed from the paper;
- In Figure 4 mentions a series of times for heat treatment operations. These times may be valid for a certain piece and are not some, in general. Thus, a justification of these times is required, for a certain piece that is not presented in the paper;
- The metallographic structures presented in Figure 5 have a very low resolution. Also, the sampling area from the piece of specimens for which the metallographic structure is presented must be specified;
- It is specified that two steel specimens were analyzed, namely C 45 and 16MnCr5 respectively. This choice is not justified, especially since it is known that the two steels behave differently in the case of heat treatments;
- it is specified that 24 specimens were made for each steel. It is not understood from the paper how they are differentiated;
- the presented diagrams show different dependencies on the speed of a fan. The characteristics of this fan are not specified;
- it is specified in sub-item 4.2 that a car steering gears has been tested. For this part, only a diagram is presented, which shows a change in the cooling time with the fan speed. This is known, but what are the effects on the characteristics of the material in the piece?
- In most cases the figures have a very low resolution;
- the presentation of the paper does not show the novelty brought by the researches performed;
- the future research directions are some general ones, and as they are presented I don't think they will bring novelty in the field;
- the conclusions are general without highlighting the research contributions brought by this paper;
Thus, the article cannot be accepted for publication in this form and propose the rejection.
Author Response
Dear Editor and Reviewers,
The authors would like to thank the editor and reviewers very much for sending the detailed reviews. The comments sent allowed for significant raise in the scientific level of the article. The most important changes in the corrected version of the manuscript have been highlighted in red, while the grammatical and stylistic changes were not marked in order to make the document more readable. The detailed answers for 3rd reviewers’ remarks may be found in separate file.
regards,

Round 2
Reviewer 3 Report
The authors responded to the made comments in review. The article can be published in the presented form.